# Tribological Performance and Model Establishment of Self-Compensating Lubrication Film Inspired by the Functional Surfaces of *Scapharca subcrenata* Shells

Guanchen Lu [1] and Zhijie Yang [2,*]

1   School of Mechanical Science and Engineering, Huazhong University of Science and Technology, Wuhan 430074, China; luguanchen@hust.edu.cn
2   School of Mechanical and Electronic Engineering, Wuhan University of Technology, 122 Luoshi Road, Wuhan 430070, China
*   Correspondence: yangzhijie@whut.edu.cn

**Abstract:** Composite surface structures inspired by the functional surface of *Scapharca subcrenata* shells can improve the tribological properties effectively, composed of the ordered "U"- shape micro-grooves and solid lubricant Sn-3.0Ag-0.5Cu (SAC305) alloys. A series of wear tests were conducted to further investigate the formation characteristics of the self-compensating lubrication film, and then the mathematical model of the spreading tribofilm could be proposed. The results showed that the appropriate surface texturing parameters (NBCSS-28) had a great effect on the formation of the self-compensating lubrication film, which exhibited a lower friction coefficient (0.386) and wear volume (0.682 mm$^3$) than the other NBCSS samples. The tribofilm, with a thickness of a few microns, was deposited on the contact surface after the wear tests. The interfacial reactants (the Ni/Ni$_3$Sn$_2$ interface) of the SAC305 alloys, and Ni$_3$Al alloys confirmed by the wetting experiments and the thermogravimetric analysis, could promote the deposition and diffusion of the tribofilm during the sliding process. Hence, distinguishable layered structures could be observed on the fractured surfaces of the NBCSS samples. Moreover, the formation process of the tribofilm exhibited an obvious relationship with the reduction in the dynamic friction coefficient. The tribofilm formation model was proposed by the accumulation behaviors of the spreading tribofilm randomly in the form of multiple discrete irregular film shapes on the worn surface, which could predict the formation characteristics of the self-compensating lubrication film to improve the optimization design of the parameters.

**Keywords:** tribological performance; model establishment; self-compensating lubrication film; biological characteristics

## 1. Introduction

The regulation design of the interfacial tribological characteristics effectively reduced the wear behaviors and prolonged the service life of the frictional parts, which could be inspired by nature. Recent research has made great progress in designing and preparing the structure of the frictional parts by imitating the biological composite surface to achieve functions [1–8] such as water collection, anti-fouling, wear resistance, and super-hydrophobicity. Inspired by the surface morphology of *Sarracenia trichomes* [9], the layered slotted surfaces were fabricated by Wan et al., and they exhibited a better fog collection ability than flat surfaces at low temperatures. Cui et al. [10] also made the conclusion that the transporting behaviors and cutting performance of the bio-inspired composite surface structure presented better anti-friction and wear resistance properties. The friction components with bio-inspired surface micro-textures, such as cylinder liners, sliders, bearings, and cutting tools, exhibited outstanding tribological performance in fluid lubrication. However, the wear resistance of surface micro-textures deteriorated rapidly under severe lubrication

conditions, such as dry friction or boundary lubrication. Huang et al. [11] proposed U-shaped structures of the shark-skin bio-inspired riblets produced by laser-assisted belt grinding processing, which used titanium alloy blades. Lu et al. [12] studied the adhesive contact of gecko-inspired groove-like textured surfaces, which exhibited excellent anti-adhesive properties compared with the flat surface. Shi et al. [13] designed the surface micro-textures inspired by the tree-frog foot on AISI4140. The bionic hexagonal texture on AISI4140 reduced the friction coefficient and wear rate by 67.93% and 42.19% compared to the untextured surface. The different surface morphologies had an important influence on friction reduction and wear resistance properties.

Recently, the cooperation effects of surface textures and solid lubricant coatings have received increasing attention from researchers [14–16]. The main factors for enhancing the wear resistance properties of bio-inspired surface composite structures were focused on the bio-inspired surface morphology and interfacial tribological characteristics. Fratzl et al. [16,17] created a unique and orderly three-dimensional pattern arranged by the soft and hard materials at a certain ratio, which was inspired by hierarchical materials in nature. The composite structures not only exhibited better tribological performance than single factors, such as surface texturing or coatings, but they also had the capacity to withstand larger loads in the poorer environment [18,19]. The coatings, such as soft metals, $Ti_3SiC_2$, and $MoS_2$, were confirmed to provide effective self-repair or self-compensating lubrication triggered by wear behaviors. Zhai et al. [20] made a series of wear tests to manifest the self-healing behavior on the surface of nanocrystalline nickel aluminum bronze/$Ti_3SiC_2$ composites during fretting wear, attributed to the simultaneous decomposition and oxidation of $Ti_3SiC_2$. Liu et al. [21] fabricated micro-poles, inspired by the biological characteristics on the surface of M50 steel. The results indicated that the volume expansion behaviors were caused by Sn and Cu oxidation to self-heal the damaged surface. Huang et al. [22,23] researched the self-repairing behaviors of SnAgCu triggered by wear. The hard phase of nano-TiC promoted the formation of spherical particles, which converted the contact forms of the lubricants and repaired the micro-grooves and furrows on the worn surface. Meanwhile, the optimized bionic texture parameters were obtained by response surface methodology (RSM).

In response to the industrial demands for sliding components with anti-friction and wear resistance, the relevant basic research was conducted by our team. Based on the shell-like composite structures that are bio-inspired by *Scapharca subcrenata*, the shell-like surface composite structures not only reduced the friction but also suppressed the friction-induced vibration and noise [24–26]. According to the previous research, tribofilm can be formed between the friction pairs. The morphological characteristics and thickness of the tribofilm have a significant influence on the tribological properties. However, the research on the effectiveness and responsiveness of the tribofilm formed on the friction interface has not been sufficiently investigated. The research on the process and roles of self-compensating lubrication behaviors still exhibits a great challenge.

In this paper, the tribological performance and model establishment of a self-compensating lubrication film inspired by the functional surfaces of *Scapharca subcrenata* shells are further discussed. $Ni_3Al$ alloys, used as friction component materials, are needed to improve wear resistance under heavy loads and a lack of lubrication [27–29]. Hence, the $Ni_3Al$ matrix composite surface structure inspired by *Scapharca subcrenata* shells (NBCSS) was prepared. A series of wear tests were conducted to investigate the effects of the surface texturing parameters on the formation of a self-compensating lubrication film. The relationships between the growth of the tribofilm and the change in the friction coefficient are discussed. The mathematical model of the self-compensating lubrication film and dynamic friction coefficient were established, which could provide guidance for designing the surface texturing parameters of functional surfaces by *Scapharca subcrenata* shells.

## 2. Experimental Details and Methods

The excellent performance of the biological surfaces did not rely on a single factor acting independently, but it was based on the interaction of different systems to achieve excellent performance. Inspired by the surface structure characteristics of *Scapharca subcrenata* shells [6,30,31], the non-smooth surface structure enhanced the ability of the shell to resist the impact of gravel, and the fluffy layers that covered the shell surface could reasonably provide better lubrication performance. *Scapharca subcrenata* is the Latin scientific name for one of the shell types we studied. The surface characteristics are shown in Figure 1a.

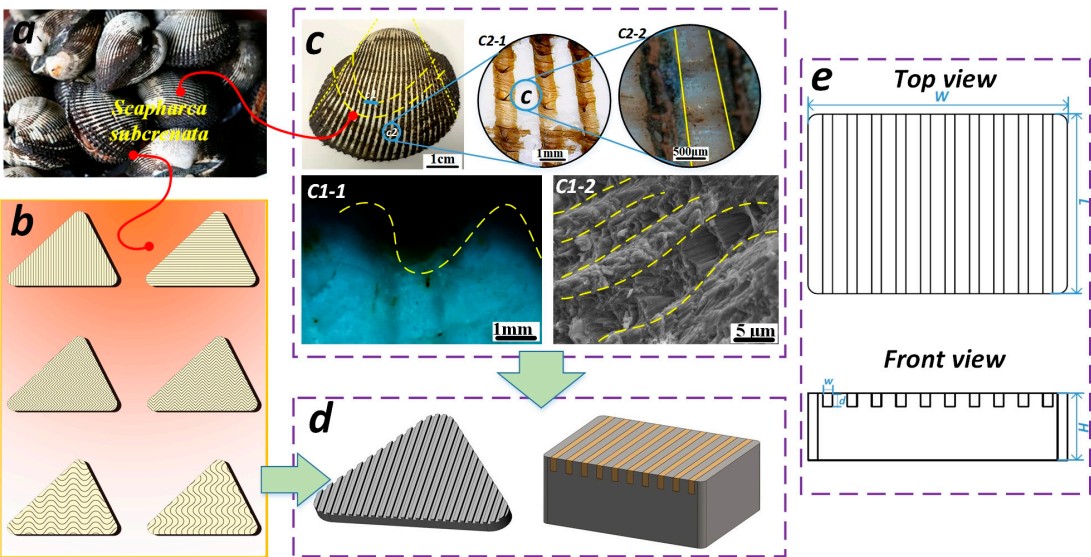

**Figure 1.** (**a**) Macro-morphologies of *Scapharca subcrenata*: (**b**) Design samples of surface micro-texture structure. (**c**) Micro-structures of *Scapharca subcrenata*: (**c1-1**) Optical morphology of shell cross-section. (**c1-2**) SEM image of fractured surface of shell. (**c2-1**) Surface morphology of shell. (**c2-2**) Local magnification of area *c*. (**d**) Composite surface structures inspired by functional surface of *Scapharca subcrenata* shell. (**e**) The dimensional views of surface textures developed on samples.

### 2.1. Confirmation of Bio-Inspired Shell-like Composite Structures

The shells used in this experiment were *Scapharca subcrenata* shells from the base of aquaculture in the Fujian province, as shown in Figure 1. The shells were rinsed with distilled water, and the debris was removed using a toothbrush. The shells were separated using a stainless steel knife, and then the closed shell muscle and soft tissue of the inner cavity were removed. Finally, the shells were washed with distilled water again and dried in an air environment. Top of the shell was facing upward and placed flat on the table, and the farthest points from the top to the edge of the shell were connected by two straight lines, which were deemed the growth lines. The macroscopic morphology of the *Scapharca subcrenata* shell showed a triangular shape. The shell surface presented multiple micro-texture structures, and the shell surface micro-grooves were inlaid with villi. The spacing of surface micro-textures on the shell surface was about 0.2–0.5 mm, and the depth was about 0.3–0.6 mm, which was measured by optical microscopy. It was observed by scanning electron microscopy that the shell matrix growth was in the form of layer-by-layer stacking, which could greatly increase the tensile stress. The composite surface structure composed of the surface micro-textures and villi had undergone a natural selection process and exhibited an excellent ability to interact with the natural environment.

### 2.2. Sample Preparation of Bio-Composite Surface Structure

The sample preparation procedure and sequence of the surface composite structure are shown in Figure 2. The materials of the composite surface structure were $Ni_3Al$ pre-alloy powders (see Figure 2a), which were composed of available Ni with 8.10 wt.% Al, 5.23 wt.%

Cr, 7.02 wt.% Mo, 0.13 wt.% Zr, and the solid lubricant Sn3.0Ag0.5Cu (SAC305) pre-alloy powders. The XRD patterns of $Ni_3Al$ matrix pre-alloy powders and SAC305 pre-alloy powders are shown in Figure 2(a1,d1), respectively. Their purity could be confirmed. Table 1 lists the mechanical and physical properties of the $Ni_3Al$ alloys and SAC305 alloys. The sample preparation procedure was as follows: (1) The multi-layer structure preparation process of the $Ni_3Al$ matrix has been proven to improve wear resistance performance. The $Ni_3Al$ alloys were prepared using layer-to-layer laser melting deposition (LMD) technology (see Figure 2b). The LMDed samples were cut into 20 mm × 10 mm × 6 mm and used for preparing the shell-like surface composite structure. The LMDed samples were polished with metallographic sandpaper, and then cleaned in acetone in an ultrasonic cleaner. (2) The micro-scale surface textures were prepared by the micro-EDM technology (see Figure 2c). The precision of the micro-EDM could reach 100 μm. (3) The solid lubricant SAC305 pre-alloy powders (see Figure 2d) were filled into the surface micro-textures and compacted at a pressure of 5 MPa. The filled sample was loaded into a hot press mold and put into a high-temperature vacuum furnace. The sintering process was set as follows: sintering temperature 450 °C, holding time 20 min, and vacuum: $1 \times 10^{-3}$ Pa, with the furnace cooling itself. The bio-composite structure samples were lightly polished, and the surface roughness was measured as *Ra* 0.1 μm using a profilometer. The composite surface structure was bio-inspired by the *Scapharca subcrenata* shells and prepared on the $Ni_3Al$ matrix. The designed sample could be termed an NBCSS sample and is shown in Figure 2e.

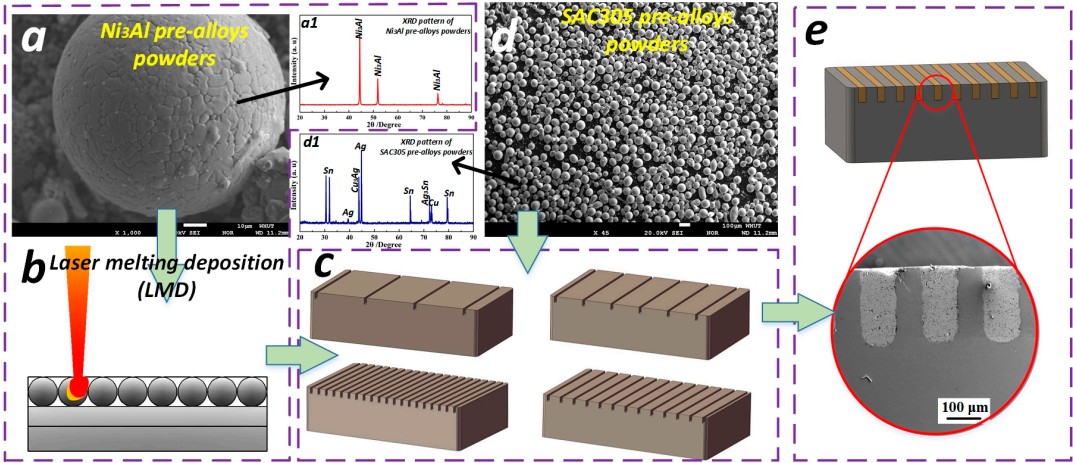

**Figure 2.** Sample preparation procedure and its sequence: (**a**) $Ni_3Al$ pre-alloyed powders: (**a1**) XRD pattern of $Ni_3Al$ pre-alloyed powders; (**b**) Schematic of $Ni_3Al$ alloys printed by laser technology; (**c**) Fabricating the samples with different surface texturing densities; (**d**) SAC305 pre-alloyed powders: (**d1**) XRD pattern of SAC305 pre-alloyed powders; (**e**) Schematic diagram and microstructure of the NBCSS sample.

**Table 1.** Mechanical and physical properties of $Ni_3Al$ alloys and SAC305 alloys [27,32].

| Materials | Micro-Hardness | Melting Point | Elastic Modulus | Thermal Conductivity | Density |
|---|---|---|---|---|---|
| $Ni_3Al$ alloys | 557.2 HV | 1390–1450 °C | 123.4 GPa | 28.85 W/(m·K) | 7.5 Kg/m$^3$ |
| SAC305 alloys | 15.2 HV | 217–220 °C | 38.7 GPa | 64 W/(m·K) | 7.37 Kg/m$^3$ |

### 2.3. Experimental Details and Materiel Characterization

In order to further investigate the lubrication function and efficiency of the NBCSS samples, a series of wear tests were conducted to further illustrate the lubricating mechanisms. The wear tests were conducted using a ball-on-disk tribometer (UMT TriboLab, Bruker Instruments, Karlsruhe, Germany) to evaluate the friction and wear behaviors of

the NBCSS samples. The high-carbon steel balls (GCr 15) with a diameter of 6 mm were chosen to the slide against the NBCSS samples, whose hardness and surface roughness were 8.0 GPa and approximately 0.1 μm, respectively. Based on the previous research on the bio-inspired surface structures, the normal load for the sliding wear tests was set to 30 N, a reciprocating sliding amplitude of 6 mm, and a frequency of 1 Hz. The sliding time was set to 3600 s, which meant that the counterpart ball was sliding on the NBCSS samples for the 3600 cycles. Every NBCSS sample with different surface texturing densities was repeated three times to ensure the repeatability of the wear tests. The lab for the wear tests adjusted to the atmospheric conditions at room temperature ($20 \pm 5\,^{\circ}\text{C}$) and $35 \pm 10\%$ relative humidity. The UMT tribometer could present dynamic friction coefficient of friction pairs in real time, which was recorded every 0.5 s by setting the parameters of the wear tests. The wear tracks, wear profiles, and surface roughness were observed and measured using a 3D optical microscope (OLYMPUS, DSX 510, Tokyo, Japan), which was used to calculate the wear volumes. The wear morphologies and elemental distributions of the NBCSS samples were analyzed using scanning electron microscopy (SU3900, HITACHI, Tokyo, Japan), with the field emission scanning electron microscope (FEI Inspect F50 FEG) equipped with energy dispersive spectrometry (EDS) equipment at an accelerating voltage of 30 kV. The phases of the pre-alloy powders were examined using X-ray diffraction (XRD, x'pert3 powder, PANalytical B.V., Tokyo, Japan) with Cu *Kα* radiation ($\lambda = 0.15406$) at a voltage of 60 kV and current of 60 mA. The fracture surfaces of the NBCSS samples were observed by field emission scanning electron microscopy (FESEM, ULTRA-PLUS-43-13, Zeiss Corporation, Oberkochen, Germany) to discuss the dynamic formation process of the self-compensating lubrication film.

## 3. Results and Discussion

### 3.1. Friction and Wear Behaviors of NBCSS Samples

In order to investigate the wear characteristics of the self-compensating lubrication film, the friction and wear behaviors of the NBCSS samples with different surface texturing densities (15, 18, 25, 28, 33, and 40%) against GCr 15 balls were investigated. The surface texturing density ($\rho$) was considered an experimental variable that could be used as the ratio of the SAC305 alloys' area to the total area of the contact surface [33]. It can be observed in Figure 3a that the surface structures of the NBCSS samples were designed to be NBCSS-15, 18, 25, 28, 33, and 40, respectively. According to the average friction coefficients of the NBCSS samples shown in Figure 3b, with the increase in the surface texturing density from 15% to 40%, the average friction coefficient initially decreased from 0.55 for NBCSS-15 to 0.35 for NBCSS-25. The average friction coefficients for NBCSS-33 and NBCSS-40 were in the range of 0.45–0.50. This indicated that the optimum ranges of the surface texturing density of the NBCSS samples during the sliding process were 25–28%. The wear scar widths of the NBCSS samples with different surface texturing densities were 531.18, 587.21, 623.91, 664.52, 771.23, and 988.43 μm, as shown in Figure 3c, which indicates that the appropriate surface texturing density exhibits excellent wear characteristics.

To evaluate the wear resistance of the NBCSS samples with different surface texturing densities, the wear scar profiles were extracted. The wear volume was obtained as follows: The samples were cleaned with acetone to remove the hydraulic oil. The wear areas were scanned sequentially using a 3D optical microscope to obtain the wear profile topographies. $Ni_3Al$ alloys (557.2 HV) are soft to compared to a commercially available high-carbon steel ball (880 HV). After the wear test, the wear volumes of the counterpart balls slid against the NBCSS samples (NBCSS-25, 28, 33), which were much lower than those of the NBCSS samples, which were $0.063\ mm^3$, $0.062\ mm^3$, and $0.073\ mm^3$, respectively. According to the previous research [25,26], the transfer film forms on the counterpart ball to prevent direct contact of the friction pairs and improve the self-compensating lubrication ability. Hence, the wear volumes of the counterpart ball were low. It could be observed that the wear volumes of the NBCSS samples with surface texturing densities of 15%, 28%, and 33% reduced rapidly. However, when the surface texturing densities of the NBCSS sample

exceeded 28%, the wear volumes of the NBCSS samples increased gradually. As shown in Figure 4a, the wear volumes of the NBCSS samples with surface texturing densities of 25%, 28%, and 33% presented lower values than the others in the wear tests, which were the minimum values of 0.754, 0.682, and 0.726 mm$^3$, respectively. The 3D wear profiles of NBCSS-25, 28, and 33 samples are shown in Figure 4b1–b3, respectively, which displays the "*V*"-, "*W*"-, or "*U*"-shaped wear profile characteristics. The obvious differences in 2D wear profiles could be seen. The surface roughness of NBCSS-25, 28, and 33 samples exhibited severe curve fluctuations. The wear depths of NBCSS-25, 28, and 33 samples were 9.94, 6.35, and 12.64 µm, respectively.

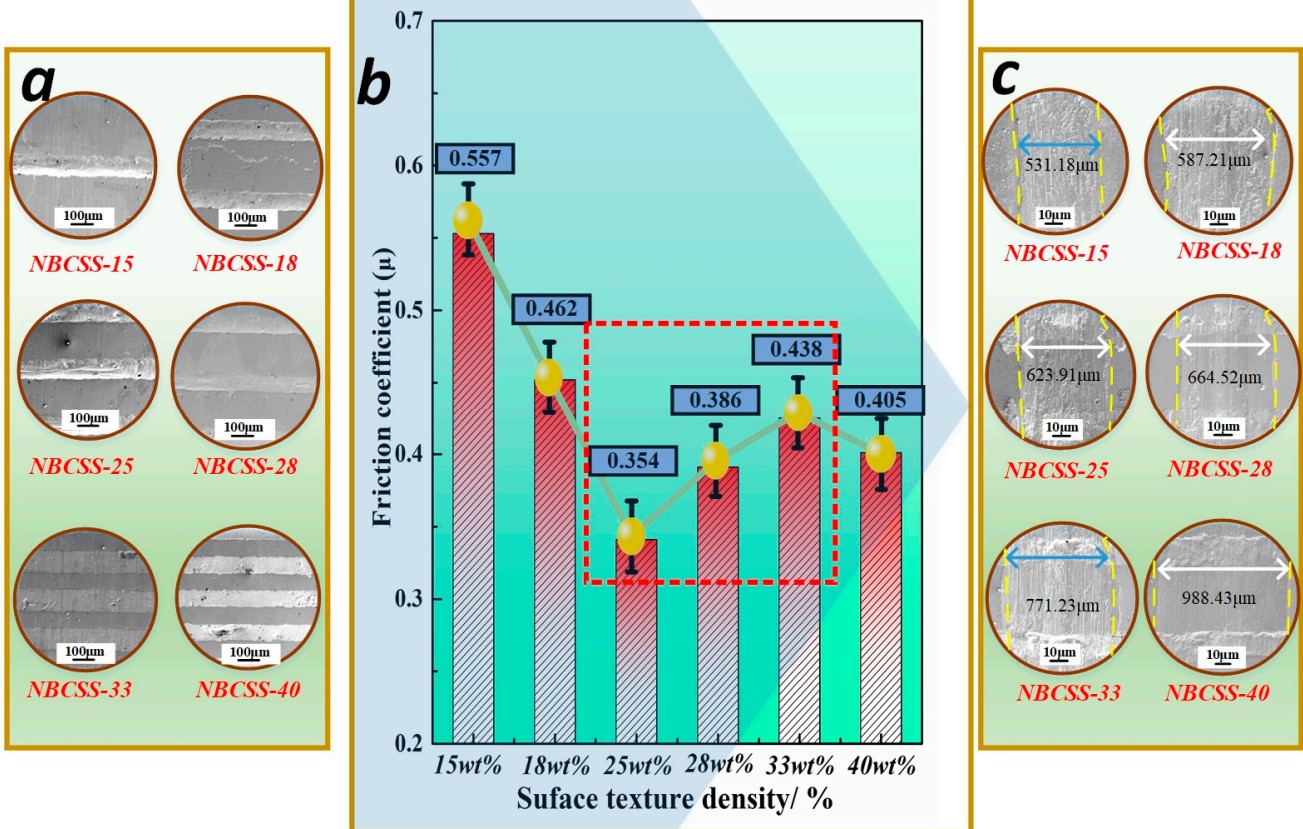

**Figure 3.** (**a**) Morphologies of the NBCSS samples with different surface texturing densities (NBCSS-15, 18, 25, 28, 33, and 40). (**b**) Average friction coefficients of NBCSS-15, 18, 25, 28, 33, and 40 during the sliding process. (**c**) Wear scar widths of the NBCSS samples with different surface texturing densities (NBCSS-15, 18, 25, 28, 33, and 40).

NBCSS-25, 28, and 33 samples presented better wear resistance performance than the others in the wear tests. In order to further investigate the effects of surface texturing density on the wear characteristics, the worn surface morphologies of the NBCSS-25, 28, and 33 samples can be observed in Figure 5. Although the edges of the surface grooves were damaged by sliding wear, the dynamic deformation of the SAC305 alloys deposited in the micro-grooves could effectively reduce the impact effect of the counterpart ball [24]. Moreover, the SAC305 alloys gradually covered the contact areas, and the tribofilm formed on the contact surface could also improve the load-bearing capacity of the NBCSS samples. However, compared with the wear morphologies of the NBCSS-25, 28, and 33 samples, the wear characteristics exhibited various differences. The worn surface morphologies of NBCSS-25 are shown in Figure 5a,b. The distinct adhesive wear could be observed on the contact surface, which was characterized by a rough surface, wear debris, and adhesion. The worn surface of NBCSS-28, shown in Figure 5c,d, presented less wear debris and a

smoother surface morphology. The tribofilm composed of SAC305 alloys was covered on the wear scar and uniformly distributed on the contact surface. The edge fracture of the surface grooves is not obvious. The worn surface morphologies of NBCSS-33 (see Figure 5e,f) exhibited visible pits and material shedding. Due to the high percentage of the solid lubricant SAC305 alloys in the composite surface structure, the contact surface of NBCSS-33 could not withstand great pressure. Meanwhile, the low hardness of the SAC305 alloys resulted in obvious material deformation and migration.

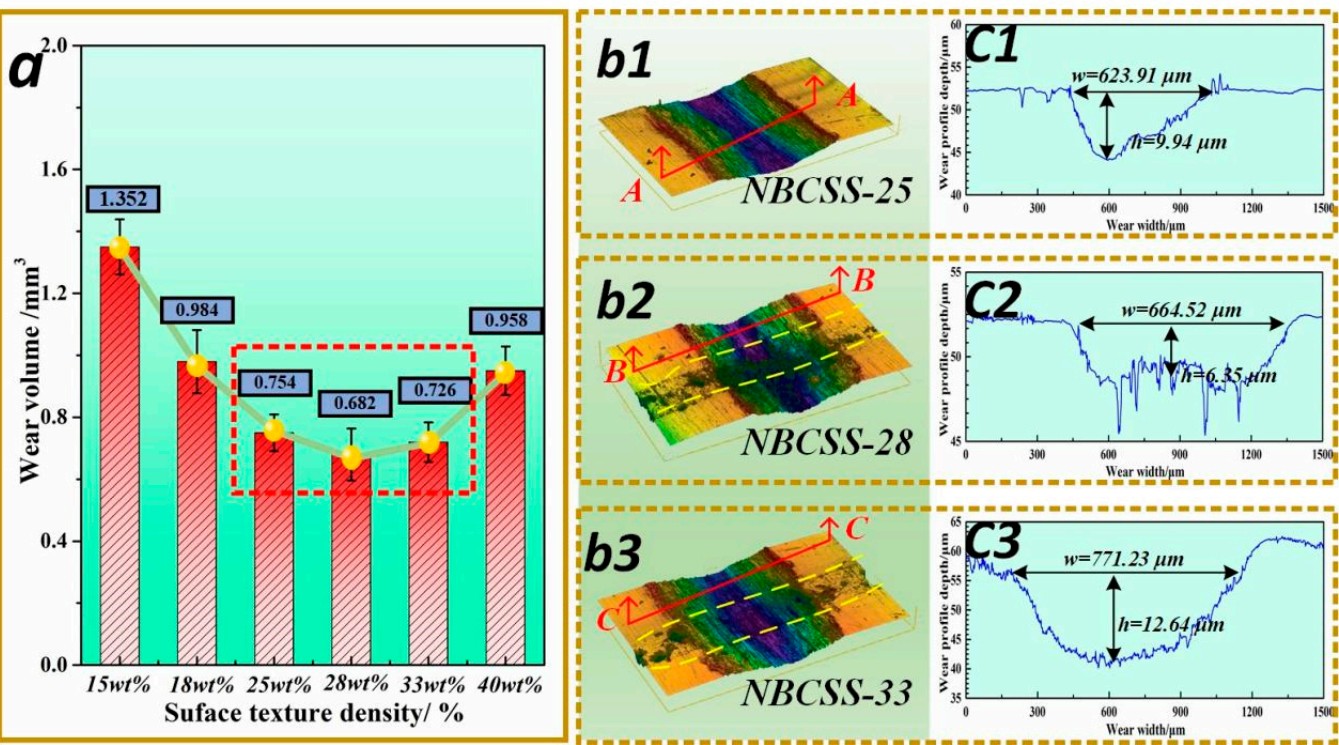

**Figure 4.** (**a**) Wear volumes of the NBCSS samples with different surface texturing densities (NBCSS-15, 18, 25, 28, 33, and 40); 3D wear profiles of NBCSS-25 (**b1**), 28 (**b2**), and 33 (**b3**) samples: 2D surface texture profiles measured by wear scar cross-section A-A, B-B, and C-C; 2D wear profiles of NBCSS-25 (**c1**), 28 (**c2**), and 33 (**c3**) samples.

In order to study the tribofilm characteristics of the NBCSS samples with different surface texturing densities, the wear scar cross-sections of the NBCSS-25, 28, and 33 samples are shown in Figure 6. The tribofilm, $Ni_3Al$ alloys, and interfacial reactants are clearly distinguishable from the wear scar fracture morphology. Because of the deformation capabilities and low hardness of SAC305 alloys, the softening behaviors of SAC305 alloys helped the formation of tribofilm during the sliding process. Based on the fracture surface characteristics of the NBCSS-25, 28, and 33 samples, the formation of the tribofilm occurred on the top layer of the contact surface, which was mainly used to reduce the friction behaviors with the counterpart ball. The chemical reaction of two materials would interdiffuse and penetrate under the action of mechanical energy. However, the self-compensating lubrication film of the NBCSS samples was not uniformly distributed like fluid lubrication. Finally, the thickness of the tribofilm was only a few microns, which exhibited good bonding with the $Ni_3Al$ matrix. The subsurface layer was made up of interface reactants to connect the tribofilm and $Ni_3Al$ alloys and prolong the service life. The tribofilm morphologies and characteristics of the NBCSS-25, 28, and 33 samples were different; the thicknesses of the NBCSS-25, 28, and 33 samples were 2.134 µm, 4.337 µm, and 2.945 µm, individually. The different surface texturing parameters had a great effect on the function and formation of the bionic self-compensating lubrication.

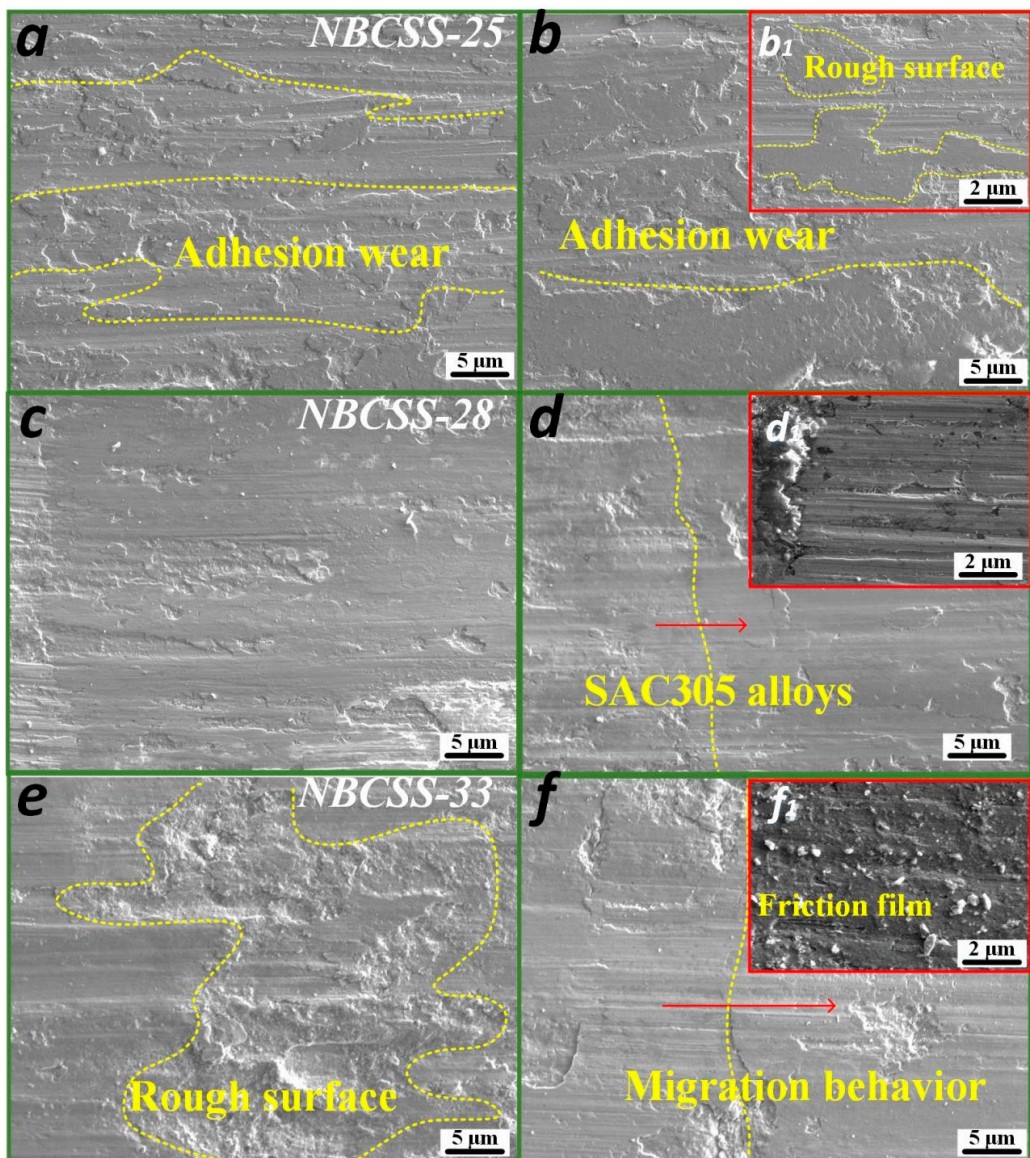

**Figure 5.** Worn surface morphologies of NBCSS-25 (**a**,**b**), 28 (**c**,**d**), and 33 (**e**,**f**) samples after wear tests, local magnification (**b1**,**d1**,**f1**) represent the wear characteristics of NBCSS-25, 28, and 33 samples.

*3.2. Solid- State Reaction Analysis of Self-Compensating Lubrication Film*

According to the analysis of the friction interface of the NBCSS samples, a layer of the tribofilm with low shear strength could be formed on the contact surface, which had the ability to adapt to the stress changes in the sliding process and reduce the direct contact between the friction pairs. The interfacial reaction characteristics had a great influence on the tribological performance of the NBCSS samples. In order to investigate the reaction mechanisms and compositions of the self-compensating lubrication film, the interaction of the main elements Ni, Al, and Sn was analyzed with a thermogravimetric analysis and wetting experiments.

Based on the Ni-Al [34], Ni-Sn [35], and Al-Sn [36] binary subsystems, the phase diagram of the Ni-Al-Sn ternary system at room temperature was established [37,38], as shown in Figure 7a. The thermogravimetric and differential thermal analysis (TG-DSC) methods were applied to the Ni-Al-Sn ternary system. Figure 7b shows the TG-DSC curves of the Ni-Al-Sn (mass ratio: 3:1:1) mixed powders heated from room temperature (25 °C) to 1000 °C. The Ni-Al-Sn composite powders showed an asymmetric exothermic peak at 217.9 °C, indicating an exothermic reaction of the tin powders and inclusions. The

Ni-Al-Sn mixed powders started to gain weight at 300 °C, illustrating that the melting of Sn powders started to react chemically with Ni and Al. With the increase in temperature, the mass of the reactants continued to increase, reaching a peak in heat absorption when the temperature was 623 °C. The nickel and tin powders were able to generate $Ni_{1-x}Sn_x$ compounds, and mechanical alloying could also be synthesized at a high temperature [37]. The crystal structure of the resulting $Ni_3Sn + Ni_3Sn_2$ phase was a mixture of the amorphous and micro-crystalline phases at all phase compositions. The phase diagram and molar ratio of the Ni-Al-Sn ternary system could infer the existence of $Ni_3Sn(r) + Ni_3Sn_2(r)$ and $Ni_3Sn(h) + Ni_3Sn_2(h)$ [39]. As the temperature continued to increase, the Aluminum started to dissolve in the $Ni_3Sn$ and $Ni_3Sn_2$ phases and Sn in the $Ni_3Al$ phase. The Ni-Al-Sn hybrid system showed no weight loss in the test of differential thermal analysis; however, the weight of the reaction product eventually increased by 26.15%.

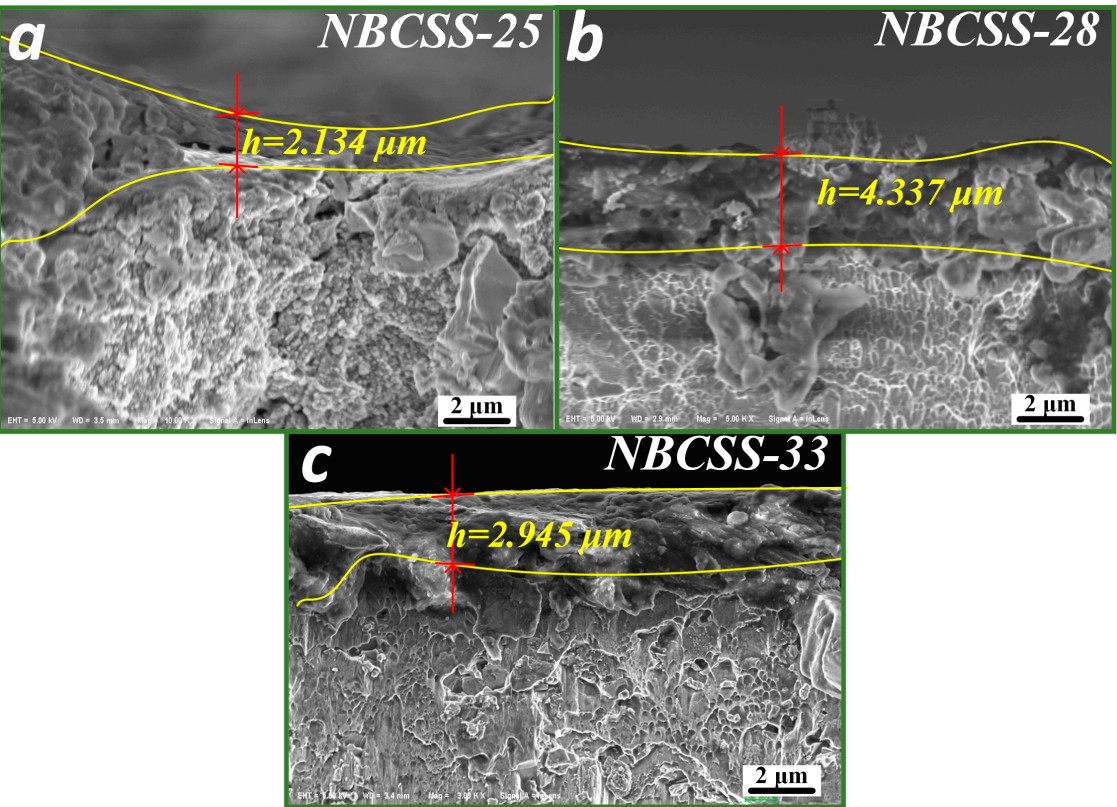

**Figure 6.** Cross-sectional FE-SEM micrographs of the NBCSS samples: NBCSS-25 sample (**a**), NBCSS-28 sample (**b**), and NBCSS-33 sample (**c**).

In order to investigate the reacting process of the SAC305 metallic powders and $Ni_3Al$ alloys, wetting experiments were conducted. The experimental procedure was as follows: the SAC305 metallic powders were placed in a mold, pressed into a rectangular shape (2 mm × 2 mm × 3 mm), and stacked on the $Ni_3Al$ alloy's surface. The sample was placed in a muffle furnace that was heated to 450 °C at a heating rate of 10 °C/min, held for 20 min, and then naturally cooled to room temperature. Figure 8 shows the polished cross-section of the interface reaction between the SAC305 alloys and the $Ni_3Al$ alloys at the different heating temperatures. The SAC305 metallic powders happened to molten at 250 °C and began to spread and inter-diffuse in the $Ni_3Al$ matrix. The thickness of the reaction layer was less than 1 μm at the contact interface (see Figure 8a). When the heating temperature increased to 450 °C, an obvious interfacial reaction layer was observed at the contact interface, and its thickness reached about 20 μm (see Figure 8b). Comparing the solid–phase contact interfaces at the two temperature conditions, their thickness and depth

were affected by the heating temperature. Moreover, a chemical reaction between tin and nickel could promote the connection between the two materials.

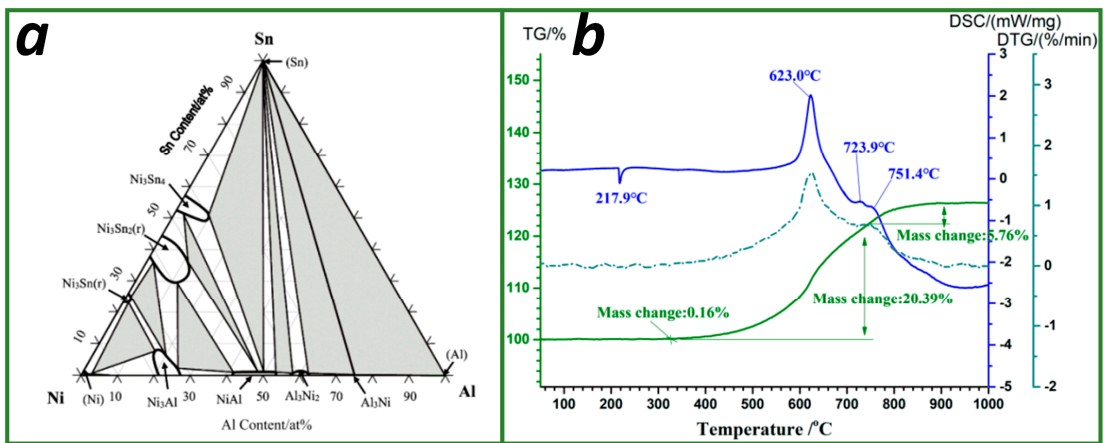

**Figure 7.** (**a**) Isothermal cross-section of Ni-Al-Sn ternary system at room temperature; (**b**) TG-DSC curve of Ni-Al-Sn ternary system from room temperature(25 °C) to 1000 °C.

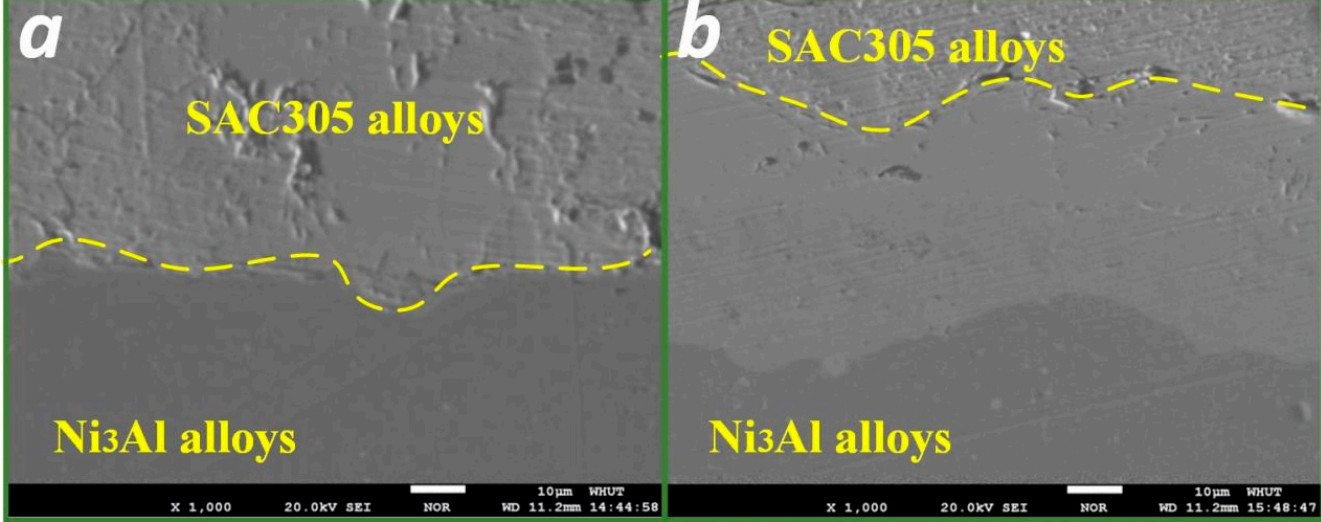

**Figure 8.** Solid- state reaction interface at the 250 °C (**a**) and 450 °C (**b**) of solid lubricant SAC305 pre-alloy powders placed on Ni$_3$Al alloys.

The interfacial elements of SAC305 alloys and the Ni$_3$Al matrix at 450 °C were analyzed using EDS analysis, as shown in Figure 9. It was found that the interfacial features showed a clear gradient laminar structure, scanning from the SAC305 alloys to Ni$_3$Al matrix. The interfacial reaction color also became darker from left to right. According to the curves of the element distribution of nickel, tin, silver, copper, and aluminum, as shown in Figure 9b, the elemental content of the tin gradually decreased while the elemental content of the nickel sharply increased, showing the diffusion of the tin element into the matrix. Figure 9c exhibited the change in the elemental content at the interface between the SAC305 alloys and the Ni$_3$Al matrix. In the initial stage, only an interfacial reaction layer was formed at the interface. The Ni$_3$Sn$_2$ compounds were formed at the Ni/Sn interface when the SAC305 alloys were molten and bonded onto the Ni$_3$Al alloy's surface at 250 °C. Then, the new reaction products were formed by extending the bonding time or improving the heating temperature, which observed that the original reaction products were no longer in local equilibrium [40,41]. When the temperature increased to 450 °C, a Ni$_3$Sn layer could be observed between the Ni$_3$Sn$_2$ and Ni$_3$Al substrates and diffuse into the Ni$_3$Al substrate.

As shown in Figure 9d, aggregation and diffusion behaviors of Ag were also found; the metal compound $Ag_3Sn$ would be replaced by the Ni base into Ag and $Ni_3Sn_2$ during the thermal process. Moreover, the $Ni_3Sn$ phase continued to be formed at the $Ni/Ni_3Sn_2$ interface, which ensured good connectivity at high temperatures.

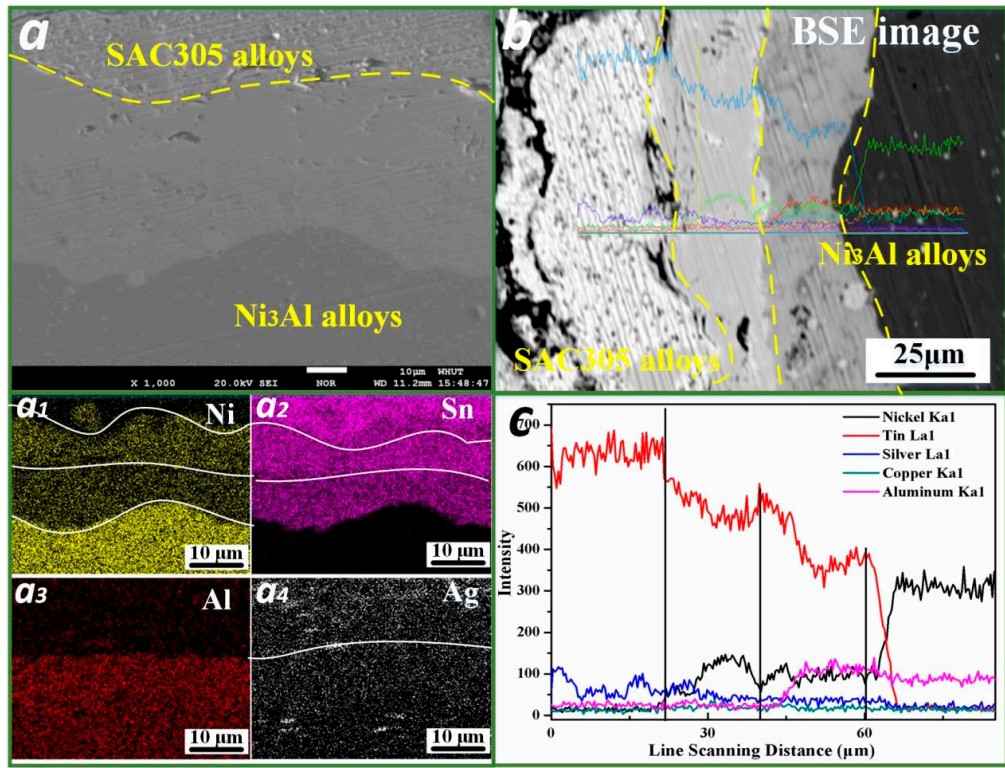

**Figure 9.** (**a**) Solid-state reaction interface at 450 °C: EDS map analysis results of interfacial reaction characteristics, Ni (**a1**), Sn (**a2**), Al (**a3**), and Ag (**a4**). (**b**) BSE image of interfacial reaction characteristics. (**c**) Line scanning analyzed from (**b**).

According to the results of the thermogravimetric analysis and wetting experiments, the diffusion of tin and nickel at the friction interface was bidirectional, under the influence of the chemical potential energy. The chemical potential energy of Ni (428 kJ/mol) was higher than that of Sn (303 kJ/mol). The driving force of the mutual diffusion between Ni and Sn was provided by the chemical potential energy difference. And the Sn elements with high chemical potential energy were more likely to diffuse into the Ni region with low chemical potential energy. The frictional heat and electron collisions helped decompose the active atoms of the SAC305 alloys. Meanwhile, the defects, such as holes and dislocations, on the wear surface could also promote the diffusion of the SAC305 alloy atoms.

### 3.3. Dynamic Formation of Self-Compensating Lubrication Film

The schematic diagrams of NBCSS-15 and NBCSS-28 are shown in Figure 10a,b; they represent the wear characteristics of the NBCSS samples with low surface texturing density and high surface texturing density, respectively. According to the wear surface morphology of NBCSS-15 in Figure 10(a1), the shear fracture of the adhesion node occurred on the friction subsurface, and abrasive chips or material migration occurred during the dry sliding process. The friction subsurface was subjected to considerable pressure and relative motion, and the worn surface showed significant surface damage. Then, the contact interface would produce local elastic–plastic deformation. The contact area formed a bump structure and a material spalling phenomenon after the repeated sliding of the counterpart ball. The phenomenon of obvious plow grooves and abrasive wear debris accumulation could be observed clearly, causing increasingly serious wear (Figure 10(a2)). According

to the wear morphologies of NBCSS-28, shown in Figure 10(b1,b2), the film-like materials adhered onto the contact surface, indicating the migration and movement of the SAC305 alloys. Therefore, the adhesive behaviors promoted the migration of the solid lubricant SAC305 alloys and the tribofilm formed on the contact surface of NBCSS-28.

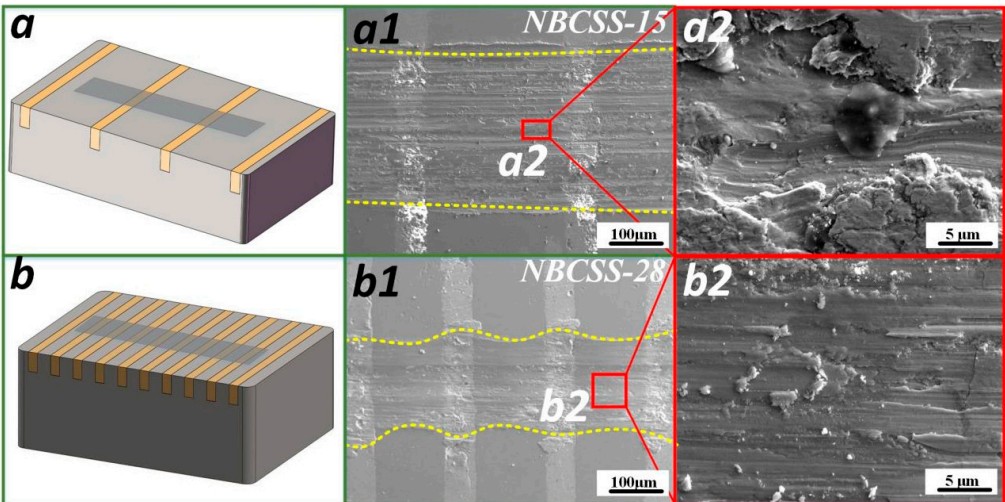

**Figure 10.** The schematic diagrams represent for the samples with low surface texturing density (**a**) and high surface texturing density (**b**), respectively; the wear scar morphology (**a1**) of NBCSS-15: local magnification (**a2**) of worn surface morphology: yellow lines represent for the width of wear track; the wear scar morphology (**b1**) of NBCSS-28: local magnification (**b2**) of worn surface morphology: yellow lines represent for the width of wear track.

The tribofilm formation characteristics of NBCSS-28 were further investigated. According to the wear scar morphology of NBCSS-28, shown in Figure 11a, the shape of the wear scar was relatively uniform, and rough surface characteristics were observed on the SAC305 alloy areas. The EDS map analysis results (Figure 11(a1–a4)) of the NBCSS-28 wear scar presented that the Ni$_3$Al alloys' wear scar was covered by a large amount of tin and silver elements, indicating the release of the SAC305 alloys. The polished fractured surface of the NBCSS-28 wear scar is shown in Figure 11b. The distinguishable layered structure was observed to adhere to the worn surface, which confirmed that tribofilm, with a certain thickness, was deposited on the contact interface. Moreover, the EDS map analysis results (Figure 11(b1–b4)) also demonstrated the diffusion of nickel and tin elements. Hence, the friction behaviors effectively promoted the chemical reaction temperature and reaction rate between the SAC305 alloys and the Ni$_3$Al alloys. The solid- state reaction of the self-compensating lubrication film could have occurred during the sliding process.

Cooperation methods of theoretical modeling and experimental measurements were used to discuss the dynamic formation process of a self-compensating lubrication film of NBCSS-28 at the friction interface. The schematic diagram of the formation process of the self-compensating lubrication film is shown in Figure 12(a1–a4). The solid lubricant SAC305 alloys overflowed and migrated to the Ni$_3$Al alloy areas when the counterpart ball slid across the solid lubricant SAC305 alloy areas. The film-like SAC305 alloys were randomly distributed on the Ni$_3$Al alloy areas under the influence of the tangential force, and the areas of the SAC305 alloys on the Ni$_3$Al alloy areas gradually increased until the generation of a complete tribofilm was achieved. The measurement of the tribofilm on the wear scar using the optical profilometer imaging technique showed the surface morphologies at sliding periods of 800 s, 1600 s, 2700 s, and 3600 s, respectively, as shown in Figure 12(b1–b4). The tribofilm with a certain thickness could be observed clearly. In order to measure the thickness of the tribofilm, an organic solvent was used to remove the SAC305 film. The NBCSS-28 wear scar demonstrated a significant height difference ($\Delta h$) before and after the cleaning behaviors using the organic solvent, which could obtain the average

thickness of the tribofilm formed on the contact surface. Based on the measurements and calculations of the surface profile, as shown in Figure 12(c1–c4), the film thicknesses were $0.427 \pm 0.025$ μm, $1.018 \pm 0.034$ μm, $1.948 \pm 0.019$ μm, and $3.136 \pm 0.028$ μm, respectively, at sliding times of 800 s, 1600 s, 2700 s, and 3600 s. The dynamic friction coefficients at the four sliding stages of NBCSS-28 are shown in Figure 12d. The wear tests of the NBCSS-28 against the GCr-15 ball were repeated three times. The dynamic friction coefficient increased rapidly in the first stage. The contact interface of the friction pair needed to overcome the influences of the surface microprotrusions, dust, and particles at the beginning of dry friction sliding [42,43], which resulted in a sharp increase in the friction coefficient. The dynamic friction coefficient started to decline during the second and third stages, which might be attributed to the role of the solid lubricants, whereby a small amount of the solid lubricant SAC305 alloy migrated to the contact surface. With the reduction in the dynamic friction coefficient, the tribofilm area on the NBCSS-28 wear scar increased rapidly. The stable friction coefficient finally reached below 0.35 at the fourth stage. This indicated that the combination of solid lubricant compositions and surface texturing parameters was likely to be the key factor in reducing the frictional force. Based on the above analysis, the self-compensating lubrication film exhibited excellent anti-friction and wear resistance performance, and the composite surface structures inspired by *Scapharca subcrenata* shells could be established to characterize the spreading process of the tribofilm. Johnson's classical theory of adhesive contact [44] proposed that the two objects, which were the rigid ball and the flexible disk, always had mutual attraction (van der Waals forces). Therefore, the fact that the counterpart ball was sliding across the $Ni_3Al$ alloys or the SAC305 alloys, respectively, presented different adhesive properties. Due to their low hardness and melting point, the SAC305 alloys were easily adsorbed on the counterpart ball and transferred to the $Ni_3Al$ alloy areas. Combined with the characteristics of the dynamic friction coefficient and the tribofilm (see Figure 12(b1–b4) and 12d), the reduction in the friction coefficient had a positive correlation with the spreading area of the self-compensating lubrication film.

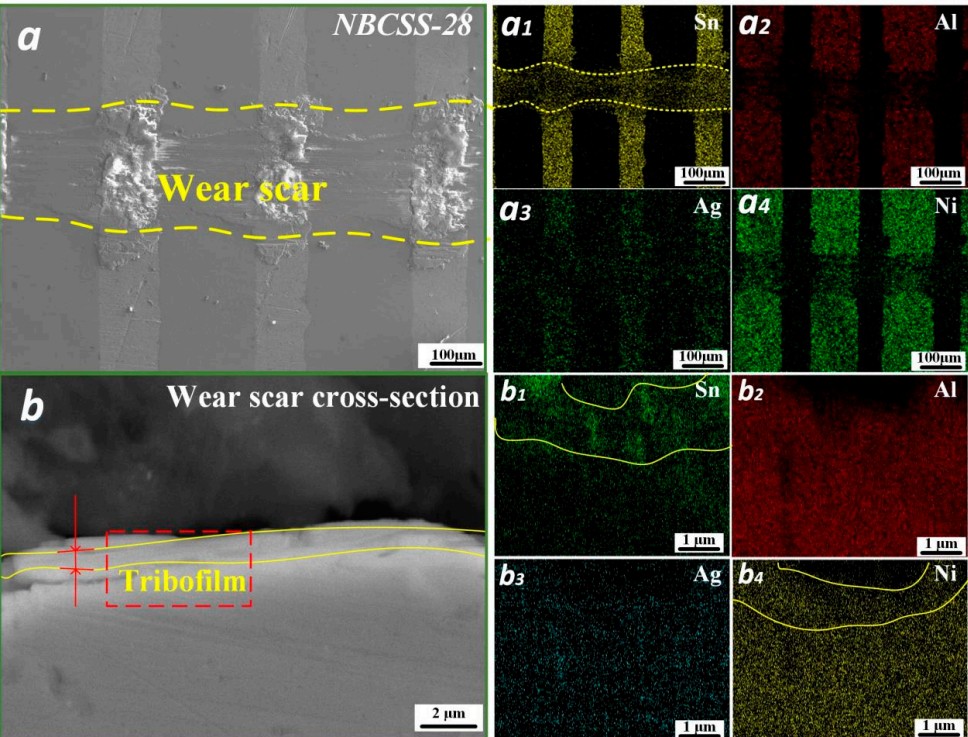

**Figure 11.** Wear scar morphology (**a**) and polished cross-section of the fracture surface (**b**) of NBCSS-28. EDS map analysis results of NBCSS-28 wear scar: (**a1**) Sn, (**a2**) Al, (**a3**) Ag, and (**a4**) Ni. EDS map analysis results of fracture surface polished cross-section of NBCSS-28: (**b1**) Sn, (**b2**) Al, (**b3**) Ag, and (**b4**) Ni.

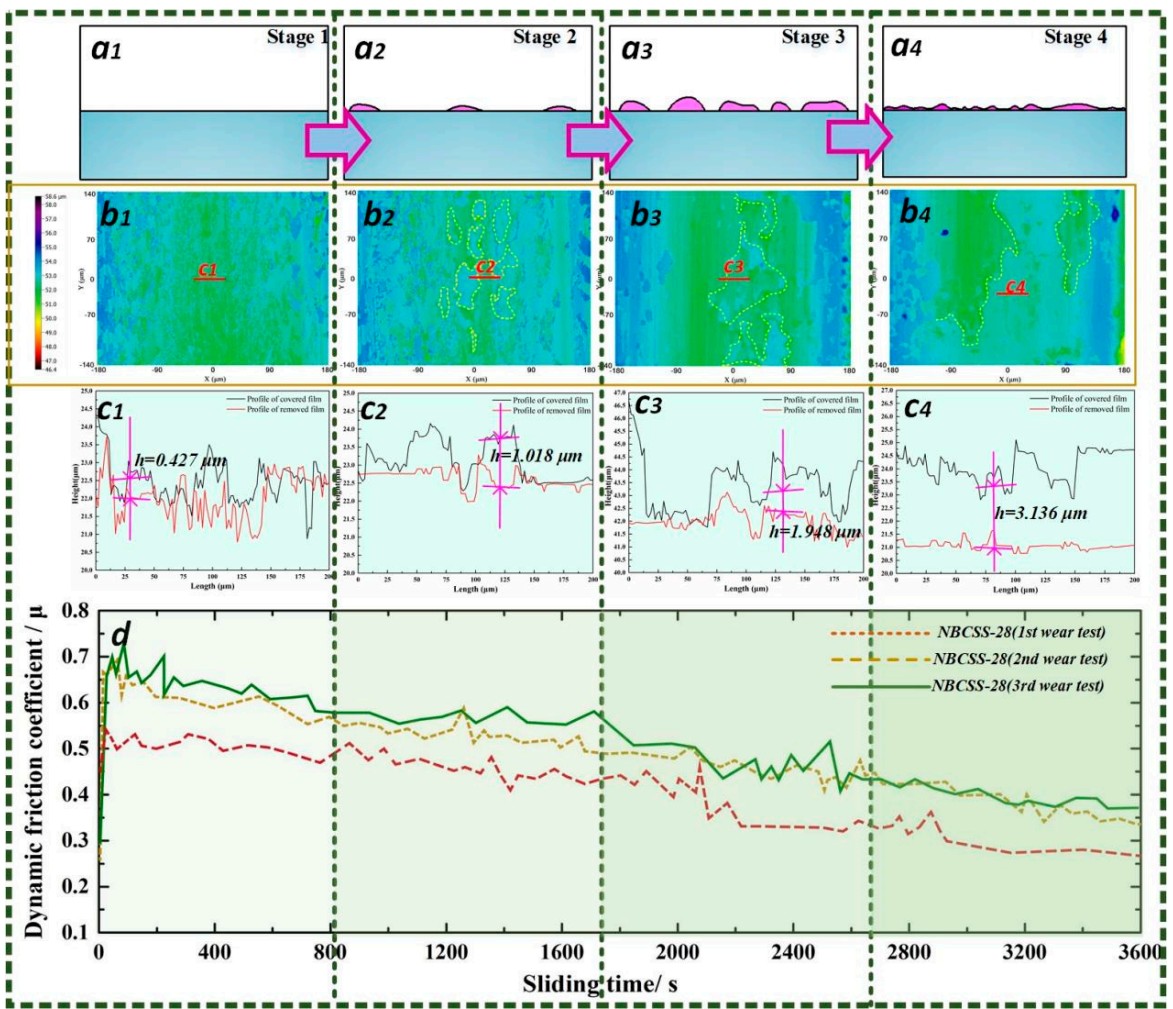

**Figure 12.** The schematic diagrams of the formation process of the self-compensating lubrication film at the four stages: (**a1**) stage 1, (**a2**) stage 2, (**a3**) stage 3, and (**a4**) stage 4; 2D wear profile imaging color maps by optical microscope at the different wear stage: (**b1**) 800 s, (**b2**) 1600 s, (**b3**) 2700 s, and (**b4**) 3600 s. thickness of self-compensating lubrication film at the four stages: (**c1**) 400 s, (**c2**) 1200 s, (**c3**) 2400 s, and (**c4**) 3600 s; (**d**) the dynamic friction coefficient of NBCSS-28 during the sliding process.

For the dynamic process of film formation of the Ni$_3$Al matrix composite surface structure inspired by the *Scapharca subcrenata* shells (see Figure 12), the tribofilm initially existed in the form of multiple discrete irregular film shapes that are randomly distributed on the worn surface. Then, it spreads constantly to the unspread area until the film formation rate gradually decreases after covering the wear surface to form a complete film. A mathematical model was developed that only considered the growing process of the self-compensating lubrication film on the Ni$_3$Al alloy's surface, ignoring the wear volume of the Ni$_3$Al alloys. If we assume that the irregular tribofilm spreads outward in a circular-like structure, the spreading area model of the self-compensating lubrication film should be expressed as a proportional relationship between the rate of spreading of the lubrication film and the incomplete covering area [45,46].

The radius and spacing of the tribofilm displaying the circular-like structure were $r_i$ and *2R*, respectively. The several tribofilms spreading on the Ni$_3$Al alloy's surface exhibited a square array. The radius of the self-compensating lubrication film grew with the increase in sliding time:

$$\frac{dr_i}{dt} = k \tag{1}$$

where $k$ was the growth constant and these circular lubrication films grew infinitely ($r \leq R$), then the lubricant coverage fraction varied with the sliding time as follows:

$$S = \frac{\pi(r_i + kt)^2}{4R^2}, r = r_i + kt \tag{2}$$

where $S$ was the fraction of the contact surface covered by the tribofilm. It was assumed that the thickness of the tribofilm covering the Ni$_3$Al alloy area was $h_{max}$, which corresponded to the final progressive thickness. Then, the average thickness expression of the tribofilm was $h_{mean} = h_{max}S$. Combined with Equation (2), it was revealed that

$$h_{mean} = h_{max}\frac{\pi(r_1 + kt)^2}{4R^2}, r \leq R \tag{3}$$

When the radius of the irregular tribofilm was $r = R_0$, the adjacent tribofilms became in contact with each other. The tribofilm began to spread to the non-spreading peripheral area, and the spreading rate of the tribofilm gradually decreased. The tribofilm was spread on the region of length $R_0$, and the maximum spread area was $R_0^2$.

$$S = \frac{4Rr\sin\alpha + (\pi - 4\alpha)r^2}{(2R)^2}, ; r = r_1 + kt \tag{4}$$

The remaining area of the tribofilm was modeled. The maximum spreading area was $\cos\alpha = R/r = \sqrt{2}/2$. When $\alpha$ was 45°, the tribofilm covered the whole contact surface. The tribofilm coverage range was $0 \leq \alpha \leq \pi/4$, $\alpha = \arccos(R/r)$.

$$S = \frac{r\sin(\arccos(R/r))}{R} + \frac{\pi r^2}{4R^2} - \frac{r^2\arccos(R/r)}{R^2} \tag{5}$$

When $r = 2^{1/2}R$ and $S = 1$, the growing behaviors of the tribofilm would stop completely, and the tribofilm reached the final progressive thickness. The film formation pattern of the SAC305 alloys on the surface of Ni$_3$Al alloys thickened gradually and became uniformly distributed, and there was a balance between the consumption and increment of the self-compensating lubrication film.

$$h_{mean} = h_{max}\frac{\pi(r_1+kt)^2}{4R^2}, r = r_1 + kt \leq R$$
$$h_{mean} = \frac{h_{max}}{4} \cdot \frac{\left(4Rr\sin(\arccos(R/r)) + \pi r^2 - 4r^2\arccos(R/r)\right)}{R^2}, R < r \leq \sqrt{2}R \tag{6}$$
$$h_{mean} = h_{max}, r \geq 2^{1/2}R$$

Although the mathematical models of the relationship between tribofilms and friction coefficients in real-world situations are more complex, the tribofilm formation model was proposed by the accumulation behaviors of the spreading tribofilm randomly in the form of multiple discrete irregular film shapes on the worn surface, which could predict the formation characteristics of the self-compensating lubrication film to improve parameter optimization design. The self-compensating lubrication film, with a small micron thickness, could be formed on the contact surface when the composite surface structure, inspired by the *Scapharca subcrenata* shells, slides against the counterpart ball. It had the great possibility of being used for bearings or other parts at high temperatures or in a vacuum environment.

## 4. Conclusions

The Ni$_3$Al matrix composite surface structure inspired by the *Scapharca subcrenata* shells (NBCSS) was prepared. A series of wear tests were conducted to investigate the effects of the surface texturing parameters on the formation of the self-compensating

lubrication film. The relationship between the growth of the tribofilm and the change in the friction coefficient was discussed. The main conclusions are as follows:

(1) The friction and wear behaviors of the NBCSS samples with different surface texturing densities (15, 18, 25, 28, 33, and 40%) against GCr 15 balls were investigated. The wear volumes of the NBCSS samples with surface texturing densities of 15%, 28%, and 33% reduced rapidly, which were the minimum values of 0.754, 0.682, and 0.726 mm$^3$, respectively. The NBCSS-25, 28, and 33 samples presented better wear resistance performance than the others in the wear tests. The appropriate surface texturing parameters (NBCSS-28) had a great effect on the formation of the self-compensating lubrication film, which exhibited a lower friction coefficient and wear volume than the other samples.

(2) These tribofilms at the contact interface grew continuously until they covered most of the surface. The wetting experiments and the thermogravimetric analysis were conducted to investigate the reacting processes of the SAC305 metallic powders and Ni$_3$Al alloys. A chemical reaction between tin and nickel could promote the connection between the two materials. It had been confirmed that the interfacial reactants of the SAC305 alloys and Ni$_3$Al alloys promoted the connectivity of the tribofilm during the sliding process.

(3) The thickness of the self-compensating lubrication film was measured by a 3D optical microscope. It exhibited an obvious relationship between the growth of tribofilm at the friction interface and the decrease in the friction coefficient. The irregular tribofilm spread outward in a circular-like structure had been assumed, and the spreading area model of the self-compensating lubrication film should be expressed as a proportional relationship between the rate of spreading of the lubrication film and the incomplete covering area, which could provide guidance for designing surface texturing parameters of functional surfaces by the *Scapharca subcrenata* shells.

**Author Contributions:** G.L.: Visualization, Methodology, Writing—original draft, Writing—review and editing. Z.Y.: Formal analysis, Validation, Investigation. All authors have read and agreed to the published version of the manuscript.

**Funding:** This research was funded by the National Natural Science Foundation of China grant number (52175169). The APC was funded by the Hubei Post-doctoral Innovation Research Position Project.

**Institutional Review Board Statement:** Not applicable.

**Informed Consent Statement:** Not applicable.

**Data Availability Statement:** Not applicable.

**Conflicts of Interest:** The authors declare no conflict of interest.

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
