# Peer review of "Tribological Performance and Model Establishment of Self-Compensating Lubrication Film Inspired by the Functional Surfaces of Scapharca subcrenata Shells"

_coatings, doi:10.3390/coatings13081399_

Round 1

Reviewer 1 Report

The authors have quantified the self lubrication and tribo film formation in bio composite bearings

Display some quantitative results in the abstract, also highlight why this work is important and how it is novel compared to already available bearings

line 35 typo superhydrophobicity

please check for grammar very carefully, in many places tenses are missing

line 96......"..................and the optimal morphology and thickness of tribo-film have not been discussed insufficiently." meaning?

There are research on multi pad bearings for self lubrication. How is this work better compared to multi pad bearings?

Author Response

Responses to Reviewer #1:

Thank you for the constructive criticisms of my manuscript. We think that the feedback resulted in valuable additions, changes, and improvements to our manuscript. According to your suggestions, the manuscript has been revised, all the changes and modifications were highlighted in Red color.

  1. line 35 typo superhydrophobicity

Thanks for your good suggestions and we have revised this part of our manuscript. We feel sorry for our poor writings.

  1. please check for grammar very carefully, in many places tenses are missing

Thank you for your given excellent comments in your busy schedule. We feel sorry for our poor writings, we do invite a friend of us who is a native English speaker to help polish our article. In this revised version, changes to our manuscript were all highlighted within the document by using red colored text. Meanwhile, based on your revised version, we have modified the other mistakes and contents for better understanding the main meanings and findings of our work. We hope the revised manuscript could be acceptable for you.

  1. line 96......"..................and the optimal morphology and thickness of tribo-film have not been discussed insufficiently." meaning?

We feel great thanks for your professional review work on our article. We have revised the sentences on Line 96 for better understanding the main meanings. “ In response to the industrial demands for sliding components with anti-friction and wear resistance, the relevant basic researches have been conducted by our team. Based on the shell- like composite structures designed and inspired by Scapharca Subcrenata, the shell- like surface composite structures not only reduce the friction, but also suppress the friction- induced vibration and noise. According to the previous researches, the tribo-film can be formed between the friction pairs. However, the researches on the effectiveness and responsiveness of the tribo-film formed on the friction interface have not been investigated insufficiently. The morphological characteristics and thickness of the tribo-film have the influences on the tribological properties. The research on the process and roles of the self- compensating lubrication behaviors still exhibited the great challenge.

  1. There are research on multi pad bearings for self lubrication. How is this work better compared to multi pad bearings?

The multi pad bearings also can reduce the wear behaviors, which attributes to the self-lubricating pad materials adhered on the inner ring and outer race. The structure seems like the coatings. The multi pad material directly determines the life of self-lubricating bearings. The multi pad material and bearing exhibit the compact structure, high bearing capacity, and easy to install and dismantle. Our research sought to improve the tribological performance by designing the surface composite structure on a Ni3Al metal matrix, in which surface micro-textures were filled by solid lubricants 96.5Sn-3.0Ag-0.5Cu alloys (SAC305). The synergistic effect of surface micro-textures and solid lubricants is a promising approach to overcome the failure of liquid lubricants in the fields of energy equipment and aerospace industry. The two composite structures have the own advantages and disadvantages and are used for the different fields respectively.

Reviewer 2 Report

Tribological performance and model establishment of self-  compensating lubrication film inspired by functional surfaces of Scapharca Subcrenata shells

While the work is overall carried out well and the review support the conclusion, there are several issues that need attention and upon addressing those issues the paper can be accepted

1-    Firstly, there are numerous typos (overtyping) throughout the manuscript, all requiring attention (the abstract has such errors).  There are several grammatical errors that needed to be corrected. I urge the authors to thoroughly go through the entire manuscript and check every line for spelling, grammar, or sentence construction-related errors as without these measures the account is unreadable. 

2-    Explain each word for first time in the beginning and used its abbreviation after that [eg: Scapharca subcrenata shells (NBCSS)]

3-      Proper conclusion outcome of all items to be presented in the manuscript

Author Response

Responses to Reviewer #2:

Thank you for the constructive criticisms of my manuscript. We think that the feedback resulted in valuable additions, changes, and improvements to our manuscript. According to your suggestions, the manuscript has been revised, all the changes and modifications were highlighted in Red color.

1- Firstly, there are numerous typos (overtyping) throughout the manuscript, all requiring attention (the abstract has such errors). There are several grammatical errors that needed to be corrected. I urge the authors to thoroughly go through the entire manuscript and check every line for spelling, grammar, or sentence construction-related errors as without these measures the account is unreadable.

Thank you for your given excellent comments in your busy schedule. We feel sorry for our poor writings, we do invite a friend of us who is a native English speaker to help polish our article. In this revised version, changes to our manuscript were all highlighted within the document by using red colored text. Meanwhile, based on your revised version, we have modified the other mistakes and contents for better understanding the main meanings and findings of our work. We hope the revised manuscript could be acceptable for you.

2- Explain each word for first time in the beginning and used its abbreviation after that [eg: Scapharca subcrenata shells (NBCSS)]

Thanks for your good suggestions and professional comments. We have supplemented the introduction of the abbreviation in our manuscript. Scapharca subcrenata is the Latin scientific name for one of the shell types we studied. The surface characteristics is shown in Figure 1(a). The composite surface structure were bio-inspired by the Scapharca subcrenata shells, and prepared on the Ni3Al matrix. The designed sample could be termed as NBCSS and was shown in Figure 2(e).

3- Proper conclusion outcome of all items to be presented in the manuscript

Thanks for your good suggestions and professional comments. We have modified the conclusions of our manuscript. The main conclusions were as follows. (1) The friction and wear behaviors of the NBCSS samples with different surface texturing densities (15, 18, 25, 28, 33 and 40%) against GCr 15 balls were investigated. The wear volumes of the NBCSS samples with surface texturing densities of 15%, 28% and 33% reduced rapidly, which were the minimum values of 0.754, 0.682, and 0.726 mm3, respectively. NBCSS-25, 28, 33 samples presented better wear resistance performance than the others in the wear tests. The appropriate surface texturing parameters (NBCSS-28) had a great affect on the formation of the self- compensating lubrication film, which exhibited the lower friction coefficient and wear volume than other samples. (2) These tribofilms at the contact interface grew continuously until it covered the most of the surface. The wetting experiments and the thermogravimetric analysis had been conducted to investigate the reacting process of the SAC305 metallic powders and Ni3Al alloys. A chemical reaction of the tin and nickel could promote the connection of the two materials. It had been confirmed that the interfacial reactants of the SAC305 alloys and Ni3Al alloys promoted the connectivity of tribofilm during the sliding process. (3) The thickness of the self-compensating lubrication film was measured by a 3D optical microscope. It exhibited an obvious relationship between the growth of tribo-film at the friction interface and the decrease of friction coefficient. The irregular tribofilm spread outward in a circular-like structure had been assumed, the spreading area model of self- compensating lubrication film should be expressed as a proportional relationship between the rate of spreading of lubrication film and the incomplete covering area, which could provide the guidance for designing the surface texturing parameters of functional surfaces by the Scapharca Subcrenata shells.

Reviewer 3 Report

I have reviewed the paper, “Tribological Performance and model establishment of self – compensating lubrication film inspired by functional surfaces of Scapharca Subcrenata Shells .Ë® This research study presents tribological properties of SAC305 alloys. After surface modification on the samples tribological studies have been conducted. Surface modification is inspired by Scapharca Subcrenata Shells. Bio inspired by Subcrenata Shells structure has been developed on the surface of samples which have micro textures.

1.     Can the authors produce three dimensional view of surface textures developed on samples.

2.     XRD of powders s not given to verify its purity.

3.     Why water has been taken as lubricant.

4.     Why high carbon steel has been chosen as counter as counter body material instead of ceramic ball.

5.     Justification for selecting normal load of 3N is not discussed.

6.     Fig.3 indicating the morphology is not clear. It should be with higher magnification.

7.     3D profile shown in Fig.4 is ok

8.      Wear track surface morphology shown in Fig.5 is ok

9.     Cross sectional view of sample shown in Fig.6 is not discussed properly and tribo film analysis interns of XRF or XRD or EDS is not shown.

10.   Wear rate counter body is not given

11.  Line scanning shown in Fig.9 is ok.

12.  What is the mean by dynamic friction coefficient.

13.  What is the reason for the assumption of equation 1

14.  Basic equation 2 is not discussed properly and how it has been achieved. Similarly equation 3 & 4 needs further clarification.

15.  Surface roughness of samples is not taken  not taken into consideration in the modeling. 

Author Response

Responses to Reviewer #3:

Thank you for the constructive criticisms of my manuscript. We think that the feedback resulted in valuable additions, changes, and improvements to our manuscript. According to your suggestions, the manuscript has been revised, all the changes and modifications were highlighted in Red color.

  1. Can the authors produce three dimensional view of surface textures developed on samples.

Thank you for your given excellent comments in your busy schedule. We benefit a lot during the revising process. We have supplemented three dimensional view of surface textures developed on samples in Figure 1 in our manuscript.

  1. XRD of powders s not given to verify its purity.

We feel great thanks for your professional review work on our article. We have supplemented the XRD patterns of Ni3Al and SAC305 pre-alloyed powders in Figure 2. The XRD patterns of Ni3Al matrix pre- alloys powders and SAC305 pre- alloys powders were shown in Figure 2(a1) and 2 (d1), respectively. Their purity could be confirmed.

  1. Why water has been taken as lubricant.

Thanks for your good suggestions and professional comments. Our research sought to improve the tribological performance by designing the surface composite structure on a Ni3Al metal matrix, in which surface micro-textures were filled by solid lubricants 96.5Sn-3.0Ag-0.5Cu alloys (SAC305). The synergistic effect of surface micro-textures and solid lubricants is a promising approach to overcome the failure of liquid lubricants in the fields of energy equipment and aerospace industry. Dry reciprocating sliding tests were conducted as a basic assessment for synergistic lubricating analysis of surface composite structure. The wear tests were conducted without lubricants.

  1. Why high carbon steel has been chosen as counter as counter body material instead of ceramic ball.

Thanks for your good suggestions and professional comments. The Ni3Al alloys (557.2 HV) is soft to compared to a commercially available high-carbon steel ball (880 HV). Hence, the friction pairs composed of Ni3Al alloys and GCr15 high carbon steel can be used for the friction components of automobile, such as metering plungers of injection systems and bearing system. The objective of this paper is to study the effect of surface texture densities on the thickness of friction film. The appropriate differences in hardness of friction pairs can avoid the excessive wear of counterpart ball that the tribological performance of Ni3Al alloys can be focused carefully.

  1. Justification for selecting normal load of 30N is not discussed.

We feel great thanks for your professional review work on our article. Based on the current research results of actual engineering application, such as the metering plungers of injection systems, the shaft and hub of propeller components of high-performance marine, the working conditions of these friction pairs are heavy-loaded, slow-speed and lack of lubrication. The wear failure of these friction pairs attributes to the adhesive wear, abrasive wear and plastic deformation. The design strategy of surface textures and solid lubricant has been provided to solve the poor-lubricating problems of friction pairs in our manuscript. Moreover, the dry sliding tests, which includes the normal load of 30N, the reciprocating distance of 6 mm, the sliding frequency of 1 Hz and the sliding time of 3600 s, are conducted as a basic assessment for a significant reference to research deeply of the actual engineering condition. Afterward, the actual engineering condition of bio-composite surface structure for the tribological application will be further studied.

  1. Fig.3 indicating the morphology is not clear. It should be with higher magnification.

Thank you for your nice comments on our article. We are sorry for our carelessness and have checked the graphs in Figure 3. We have enlarged the morphologies of NBCSS samples and rearranged the graph.

  1. 3D profile shown in Fig.4 is ok

Thanks for your good suggestions and professional comments.

  1. Wear track surface morphology shown in Fig.5 is ok

Thanks for your good suggestions and professional comments.

  1. Cross sectional view of sample shown in Fig.6 is not discussed properly and tribo film analysis interns of XRF or XRD or EDS is not shown.

Thank you for your nice comments on our article. We have supplemented the descriptions and discussions of the cross-sectional FE-SEM micrographs of the NBCSS samples (NBCSS-25, 28 and 33). Because of the deformation capabilities and low hardness of SAC305 alloys, the softening behaviors of SAC305 alloys helped the formation of tribofilm during the sliding process. Based on the fracture surface characteristics of NBCSS-25, 28, 33 samples, the formation of the tribo-film was on the top layer of the contact surface, which mainly used to reduce the friction behaviors with the counterpart ball. The chemical reaction of two materials  would interdiffuse and penetrate under the action of mechanical energy. However, the self- compensating lubrication film of the NBCSS samples had been not distributed uniformly like fluid lubrication. Finally, the thickness of the tribofilm was only a few microns, which exhibited the the good bonding with Ni3Al matrix. The subsurface layer was interface reactants to connect the tribofilm and Ni3Al alloys and prolong the service life. The tribofilm morphologies and characteristics of NBCSS-25, 28, 33 samples were different, the thicknesses of NBCSS-25, 28, 33 samples were 2.134 μm, 4.337 μm, and 2.945 μm, individually. Different surface texturing parameters had a great affect on the function and formation of the bionic self- compensating lubrication.

The thickness and morphology of tribo-film have been detected by EDS analysis. The polished fractured surface and EDS analysis results of NBCSS-28 wear scar were shown in Figure 11(b). The distinguishable layered structure could be observed to adhere on the worn surface, which confirmed that the tribofilm with the certain thickness was deposited on the contact interface. Moreover, the EDS map analysis results (Figure 11(b1-b4)) also demonstrated the diffusion of nickle and tin elements. Hence, the friction behaviors effectively promoted the chemical reaction temperature and reaction rate between the SAC305 alloys and Ni3Al alloys. The solid state reaction of the self- compensating lubrication film could be occurred during the sliding process.

  1. Wear rate counter body is not given

Thank you for your reminding. We feel sorry that we did not provide key information to analyze the counterpart ball. We have supplemented the wear characteristics of the counterpart ball. The Ni3Al alloys (557.2 HV) is soft to compared to a commercially available high-carbon steel ball (880 HV). After wear test, the wear volumes of the counterpart balls slid against the NBCSS samples (NBCSS- 25, 28, 33) were much less than those of the NBCSS samples, which was 0.063 mm3, 0.062 mm3 and 0.073 mm3, respectively. According to the previous researches [25, 26], the transfer film formed on the counterpart ball to prevent the direct contact of friction pairs and improved ability of the self-compensating lubrication ability. Hence, the wear volumes of the counterpart ball were low. We focused on the wear volumes and characteristics of NBCSS samples in our manuscript.

[25] G. Lu, W. Lu, X. Shi, et al. Tribological properties and self-compensating lubrication mechanisms of Ni3Al matrix bio-inspired shell-like composite structure. Applied Surface Science, 2022, 573: 151462.

[26] G. Lu, W. Lu, X. Shi, et al. Effects of Ni3Al matrix bio-inspired shell-like composite surface structure on interfacial tribological behaviors. Tribology International, 2022, 170: 107522.

  1. Line scanning shown in Fig.9 is ok.

Thanks for your good suggestions and professional comments.

  1. What is the mean by dynamic friction coefficient.

Thanks for your good suggestions and professional comments. Dynamic friction coefficient is the new industry standard for measuring a surface’s slip resistance. The values of dynamic friction coefficient will change over time. Understanding a material’s abrasion resistance and overall strength will help predict the durability of the composite surface structure.

  1. What is the reason for the assumption of equation 1

Thank you for your nice comments on our article. The assumption of equation 1 is as follows: For the dynamic process of film formation of the Ni3Al matrix composite surface structure inspired by the Scapharca subcrenata shells, the tribofilm initially existed in the form of multiple discrete irregular film shapes distributing randomly on the worn surface. Then it spread constantly to the unspread area until the film formation rate gradually decreases after covering the wear surface to form a complete film. If we assumed that the irregular tribofilm spread outward in a circular-like structure, the spreading area model of self- compensating lubrication film should be expressed as a proportional relationship between the rate of spreading of lubrication film and the incomplete covering area. The radius and spaced of the tribofilm displaying the circular-like structure were ri and 2R, respectively. The equation 1 can be obtained.

  1. Basic equation 2 is not discussed properly and how it has been achieved. Similarly equation 3 & 4 needs further clarification.

Thank you for your nice comments on our article. The equation 2 is obtained by the fraction of the contact surface covered by the tribofilm. It was assumed that the thickness of the tribofilm covering on the Ni3Al alloys area was the maximum thickness, which was corresponding to the final progressive thickness. The equation 3 and 4 are derived from equation 2. Combining with equation (2), the tribofilm began to spread to the non- spreading peripheral area, the spreading rate of the tribofilm gradually decreased until the two irregular tribofilms contacted each other. The remaining area of the tribofilm was modeled and the average thickness of the tribo-film can be obtained. The maximum spreading area had covered the contact surface, then the tribofilm coverage range can be obtained.

  1. Surface roughness of samples is not taken into consideration in the modeling.

Thank you for your nice comments on our article. The mathematical model was developed, which only considered the growing process of the self- compensating lubrication film on the Ni3Al alloys surface, ignoring the wear volume and surface roughness of the Ni3Al alloys. Although the mathematical models of the relationship between tribofilms and friction coefficients in real-world situations were more complex, these models could be used to predict the formation characteristics of the self-compensating lubrication film for the surface texturing parameters and a given sliding time. The more parameters should be taken into consideration to improve the accuracy of the mathematical models in the further research.

Reviewer 4 Report

The authors has presented their experimental work very nicely with very high quality images. However, this article needs to be revised very carefully for some reasons.

1. English grammar must be checked overall the manuscript. 

2. In abstract " A series of wear tests were conducted to further investigate the formation characteristics of the self- compensating lubrication film and establish the tribofilm formation model." one sentence mixing of different types of grammar. Similar to this there many others in the manuscript.

3. In the abstract experimental findings should be highlighted. Additionally, the last sentence makes the reader very confusing that indicates that even the authors are not sure about the application possibilities of their work. 

4. Introduction is unnecessarily big, it can be more concise with very required information. Some information are stated in the introduction but they hardly added value to the manuscript.

5. In the lines 90-92:  "Based on the shell- like composite structures designed and inspired by Scapharca Subcrenata, the shell- like surface composite structures not only reduce the friction, but also suppress the friction- induced vibration and noise [24–26]." seems that this statement is added only to cite the authors own articles.

6. The experimental work has been performed by using Ni3Al alloys, have the authors tried any other alloys. 

7. A results comparison with similar type of alloys, would add the value to the scientific readers.

8. The mathematical model and equations are presented at the very last, this section can have a subsection at the beginning of section 2.

9. The practical implementation possibilities of the proposed films should be mentioned.

10. Conclusion should be rewritten. Conclusion doesn't require the history of experimental procedures.

Language must be checked by professional.

Author Response

  1. English grammar must be checked overall the manuscript.

Thank you for your given excellent comments in your busy schedule. We feel sorry for our poor writings, we do invite a friend of us who is a native English speaker to help polish our article. In this revised version, changes to our manuscript were all highlighted within the document by using red colored text. Meanwhile, based on your revised version, we have modified the other mistakes and contents for better understanding the main meanings and findings of our work. We hope the revised manuscript could be acceptable for you.

  1. In abstract " A series of wear tests were conducted to further investigate the formation characteristics of the self- compensating lubrication film and establish the tribofilm formation model." one sentence mixing of different types of grammar. Similar to this there many others in the manuscript.

Thank you for your nice comments on our article. We have modified the sentence in our abstract. “A series of wear tests were conducted to further investigate the formation characteristics of the self- compensating lubrication film, and then the mathematical model of the spreading tribofilm could be proposed.” 

  1. In the abstract experimental findings should be highlighted. Additionally, the last sentence makes the reader very confusing that indicates that even the authors are not sure about the application possibilities of their work.

Thanks for your good suggestions and professional comments. We have checked and revised the abstract in our manuscript. We have supplemented the experimental findings in the abstract and the last sentence have been revised for the application possibilities of their work. Composite surface structures inspired by functional surface of Scapharca Subcrenata shell can improve the tribological properties effectively, which composed of the ordered “U” shape micro-grooves and solid lubricants Sn-3.0Ag-0.5Cu(SAC305) alloys. A series of wear tests were conducted to further investigate the formation characteristics of the self- compensating lubrication film, and then the mathematical model of the spreading tribofilm could be proposed. The results showed that the appropriate surface texturing parameters (NBCSS-28) had a great affect on the formation of the self-compensating lubrication film, which exhibited the lower friction coefficient (0.386) and wear volume (0.682 mm3) than the other NBCSS samples. The tribofilm with the few microns thickness was deposited on the contact surface after wear tests. The interfacial reactants (the Ni/Ni3Sn2 interface) of the SAC305 alloys and Ni3Al alloys confirmed by the wetting experiments and the thermogravimetric analysis could promote the deposition and diffusion of tribofilm during the sliding process. Hence, the distinguishable layered structures could be observed on the fractured surfaces of the NBCSS samples. Moreover, the formation process of tribofilm exhibited an obvious relationship with the reduction of dynamic friction coefficient. The tribofilm formation model was proposed by the accumulation behaviors of the spreading tribofilm randomly in the form of multiple discrete irregular film shapes on the worn surface, which could predict the formation characteristics of the self-compensating lubrication film to improvement of parameters optimization design.

  1. Introduction is unnecessarily big, it can be more concise with very required information. Some information are stated in the introduction but they hardly added value to the manuscript.

Thank you for the constructive criticisms of my paper. According to your suggestions, we have revised the Introduction section to make the contents more concise. The introduction has been revised as follows:

The regulation design of interfacial tribological characteristics effectively reduced the wear behaviors and prolonged the service life of frictional parts, which could be inspired by nature. Recent researches had made great progress in designing and preparing the structure of the frictional parts by imitating the biological composite surface to achieve the functions [1-8], such as water collection, anti-fouling, wear resistance, and super-hydrophobicity, etc. Inspired by the surface morphology of Sarracenia trichomes [9], the layered slotted surfaces were fabricated by Wan et al., which exhibited the better fog collection ability than flat surfaces at low temperatures. Cui et al.[10] also made the conclusions that the transporting behaviors and cutting performance of the bio-inspired composite surface structure presented the better anti- friction and wear resistance properties. The friction components with bio- inspired surface micro-textures, such as cylinder liners, sliders, bearings, and cutting tools, exhibited the outstanding tribological performance in fluid lubrication. However, the wear resistance of surface micro- textures deteriorated rapidly under severe lubrication conditions, such as dry friction or boundary lubrication. Huang et al.[11] proposed the U-shaped structures of the shark-skin bio-inspired riblets produced by laser assisted belt grinding processing, which used for the titanium alloy blades. Lu et al.[12] studied the adhesive contact of gecko-inspired groove-like textured surfaces, which exhibited excellent anti-adhesive properties compared with the flat surface. Shi et al.[13] designed the surface micro-textures inspired by the tree-fog foot on AISI4140. The bionic hexagonal texture on AISI4140 reduced the friction coefficient and wear rate by 67.93% and 42.19% compared to the untextured surface. The different surface morphologies had an important influence on friction reduction and wear resistance properties.

Recently, the cooperation effects of the surface textures and solid lubricant coatings had received increasing attention from researchers [14–16]. The main factors for enhancing the wear resistance properties of bio- inspired surface composite structures were focused on the bio-inspired surface morhpology and interifical tribological characteristics. Fratzl et al.[16,17] created an unique and orderly three- dimensional pattern arranged by the soft and hard materials at a certain ratio, which was inspired by the Nature’s hierarchical materials. The composite structures not only exhibited the better tribological performance than the single factors, such as surface texturing or coatings, but also could have the capacity of withstanding the larger loads in the poor environment [18,19]. The coatings, such as soft metals, Ti3SiC2, MoS2, etc., had been confirmed that provide an effective self-repair or self- compensating lubrication triggered by wear behaviors. Zhai et al.[20] made the series of wear tests to manifest the self-healing behavior on the surface of nanocrystalline nickel aluminum bronze/Ti3SiC2 composites during the fretting wear, which attributed to the simultaneous decomposition and oxidation of Ti3SiC2. Liu et al.[21] fabricated the micro- poles inspired by biological characteristics on the surface of M50 steel. The results indicated that the volume expansion behaviors were caused by Sn and Cu oxidation to self-heal the surface damaged. Huang et al.[22,23] made the research on the self-repairing behaviors of SnAgCu triggered by wear. The hard phase of nano-TiC promoted the formation of spherical particles, which converted contact forms of the lubricants and repaired micro-grooves and furrows on worn surface. Meanwhile, the optimized bionic texture parameters were obtained by response surface methodology (RSM).

  1. In the lines 90-92: "Based on the shell- like composite structures designed and inspired by Scapharca Subcrenata, the shell- like surface composite structures not only reduce the friction, but also suppress the friction- induced vibration and noise [24–26]." seems that this statement is added only to cite the authors own articles.

Thank you for your nice comments on our article. In the lines 90-92: "Based on the shell- like composite structures designed and inspired by Scapharca Subcrenata, the shell- like surface composite structures not only reduce the friction, but also suppress the friction- induced vibration and noise [24-26]." mainly introduce the research work of our team. According to the previous researches, the tribo-film can be formed between the friction pairs. The morphological characteristics and thickness of the tribo-film have the influences on the tribological properties. However, the researches on the effectiveness and responsiveness of the tribo-film formed on the friction interface have not been investigated insufficiently. The research on the process and roles of the self- compensating lubrication behaviors still exhibited the great challenge. Tribological performance and model establishment of self- compensating lubrication film inspired by functional surfaces of Scapharca Subcrenata shells have been investigated in our manuscript.

[24] G. Lu, W. Lu, X. Shi, et al. Tribological properties and self-compensating lubrication mechanisms of Ni3Al matrix bio-inspired shell-like composite structure. Applied Surface Science, 2022, 573: 151462.

[25] G. Lu, X. Shi, J. Zhang, et al. Effects of surface composite structure with micro-grooves and Sn-Ag-Cu on reducing friction and wear of Ni3Al alloys. Surface and Coatings Technology, 2020, 387: 125540

[26] G. Lu, W. Lu, X. Shi, et al. Effects of Ni3Al matrix bio-inspired shell-like composite surface structure on interfacial tribological behaviors. Tribology International, 2022, 170: 107522.

  1. The experimental work has been performed by using Ni3Al alloys, have the authors tried any other alloys.

Thanks for your good suggestions and professional comments. The manuscript mainly investigates the effects of the surface texturing parameters on the formation of the self- compensating lubrication film. The relationship between the growth of the tribofilm and the change of the dynamic friction coefficients has been proposed. Meanwhile, our team will continue study the other alloys later.

  1. A results comparison with similar type of alloys, would add the value to the scientific readers.

Thanks for your nice comments on our article. It is a good suggestion for us. We will make the plans to study the effects on the tribofilm formed on the contact surface of the similar types of alloys further.

  1. The mathematical model and equations are presented at the very last, this section can have a subsection at the beginning of section 2.

Thank you for your nice comments on our article. The mathematical model and equations are proposed by the accumulation behaviors of the spreading tribofilm randomly in the form of multiple discrete irregular film shapes on the worn surface. Hence, the mathematical model and equations are presented at last.

  1. The practical implementation possibilities of the proposed films should be mentioned.

Thanks for your good suggestions and professional comments. We have supplemented the practical implementation possibilities of the composite surface structure inspired by the Scapharca subcrenata shells. Although the mathematical models of the relationship between tribofilms and friction coefficients in real-world situations were more complex, the tribofilm formation model was proposed by the accumulation behaviors of the spreading tribofilm randomly in the form of multiple discrete irregular film shapes on the worn surface, which could predict the formation characteristics of the self-compensating lubrication film to improvement of parameters optimization design. The self- compensating lubrication film with a few micron thickness could be formed on the contact surface when the composite surface structure inspired by the Scapharca subcrenata shells slides against the counterpart ball. It had the great possibility to be used for the bearings or other parts at the high temperature or vacuum environment.

  1. Conclusion should be rewritten. Conclusion doesn't require the history of experimental procedures.

Thanks for your good suggestions and professional comments. We have modified the conclusions of our manuscript. The main conclusions were as follows. (1) The friction and wear behaviors of the NBCSS samples with different surface texturing densities (15, 18, 25, 28, 33 and 40%) against GCr 15 balls were investigated. The wear volumes of the NBCSS samples with surface texturing densities of 15%, 28% and 33% reduced rapidly, which were the minimum values of 0.754, 0.682, and 0.726 mm3, respectively. NBCSS-25, 28, 33 samples presented better wear resistance performance than the others in the wear tests. The appropriate surface texturing parameters (NBCSS-28) had a great affect on the formation of the self- compensating lubrication film, which exhibited the lower friction coefficient and wear volume than other samples. (2) These tribofilms at the contact interface grew continuously until it covered the most of the surface. The wetting experiments and the thermogravimetric analysis had been conducted to investigate the reacting process of the SAC305 metallic powders and Ni3Al alloys. A chemical reaction of the tin and nickel could promote the connection of the two materials. It had been confirmed that the interfacial reactants of the SAC305 alloys and Ni3Al alloys promoted the connectivity of tribofilm during the sliding process. (3) The thickness of the self-compensating lubrication film was measured by a 3D optical microscope. It exhibited an obvious relationship between the growth of tribo-film at the friction interface and the decrease of friction coefficient. The irregular tribofilm spread outward in a circular-like structure had been assumed, the spreading area model of self- compensating lubrication film should be expressed as a proportional relationship between the rate of spreading of lubrication film and the incomplete covering area, which could provide the guidance for designing the surface texturing parameters of functional surfaces by the Scapharca Subcrenata shells.

Round 2

Reviewer 3 Report

All suggestions implemented in the revised manuscript.

Author Response

Thanks for your good suggestions and professional comments.

Reviewer 4 Report

The authors has put their best efforts to improve the manuscript.

English must be checked before publish.